# The Importance of a Healthy Microbiome in Pregnancy and Infancy and Microbiota Treatment to Reverse Dysbiosis for Improved Health

**DOI:** 10.3390/antibiotics12111617

**Published:** 2023-11-11

**Authors:** Herbert L. DuPont, Madeleine Mary Hines Salge

**Affiliations:** 1Division of Epidemiology, Human Genetics and Environmental Sciences, School of Public Health, University of Texas, Houston, TX 77030, USA; 2Department of Internal Medicine, University of Texas McGovern Medical School, Houston, TX 77030, USA; 3Department of Medicine, Baylor College of Medicine, Houston, TX 77030, USA; 4Kelsey Research Foundation, Houston, TX 77005, USA

**Keywords:** microbiome in pregnancy, microbiome in infants, prebiotics, probiotics, vaginal seeding, fecal microbiota transplantation, hygiene theory

## Abstract

Background: The microbiome of newborn infants during the first 1000 days, influenced early on by their mothers’ microbiome health, mode of delivery and breast feeding, orchestrates the education and programming of the infant’s immune system and determines in large part the general health of the infant for years. Methods: PubMed was reviewed for maternal infant microbiome health and microbiota therapy in this setting with prebiotics, probiotics, vaginal seeding and fecal microbiota transplantation (FMT). Results: A healthy nonobese mother, vaginal delivery and strict breast feeding contribute to microbiome health in a newborn and young infant. With reduced microbiome diversity (dysbiosis) during pregnancy, cesarean delivery, prematurity, and formula feeding contribute to dysbiosis in the newborn. Microbiota therapy is an important approach to repair dysbiosis in pregnant women and their infants. Currently available probiotics can have favorable metabolic effects on mothers and infants, but these effects are variable. In research settings, reversal of infant dysbiosis can be achieved via vaginal seeding or FMT. Next generation probiotics in development should replace current probiotics and FMT. Conclusions: The most critical phase of human microbiome development is in the first 2–3 years of life. Preventing and treating dysbiosis during pregnancy and early life can have a profound effect on an infant’s later health.

## 1. Introduction

The initial exposure to a microbial world for an infant born vaginally is from the mother’s microbiota, influenced by maternal diet, level of stress, smoking history and living conditions [1]. The intestinal microbiome in the first 2–3 years of life participates in the programming and development of the gut immune system [2], important to immune reactivity and general health as well as to response to infectious organisms and vaccines resulting in protective immunity [3]. The intestinal microbiome and the immune system early in life can put infants on a long-term path to health or lead to medical and allergic disorders that can persist into adulthood [4]. 

Attempts to restore microbiomes with reduced diversity in pregnancy and infancy are being attempted by administration of prebiotics, probiotics and fecal or vaginal microbiota. Probiotics are appealing to the public, which is clear from the worldwide market for probiotics ($36 billion in 2013) [5]. Probiotics are regulated in the U.S. as dietary supplements, which do not require governmental approval before marketing. Mechanisms of probiotics have been shown to include inhibition of bacterial adherence, improved gut barrier function, antimicrobial effects, competitive inhibition of other bacteria, release of neurotransmitters, inhibition of inflammation, biofilm production and immune modulation [6,7,8].

Harnessing fecal-derived microbiota to reverse dysbiosis and improve diversity of the microbiome has been attempted for many medical disorders [9].

In this review, the authors describe pathways and practices via which microbiota gain access to the newborn, creating a microbiome pattern that participates in the programming of the early immune system, which sets the stage for future health or a pathway to disease. We furthermore discuss the opportunities to improve maternal–infant microbiome health via administration of prebiotics, probiotics or vaginal or fecal microbiota transplantation. Approaches to be taken in the development of next generation of probiotics are discussed. 

## 2. Methods

On 23 August 2023, PubMed was reviewed for microbiome in pregnancy, microbiome in infants, prebiotics, probiotics, vaginal seeding and fecal microbiota transplantation during pregnancy and in infancy. All abstracts were reviewed, and 220 papers were reviewed fully. Additional references were obtained after reviewing the papers. The focus of the review was on primary data and review articles were not selected.

## 3. The Microbiome during Pregnancy

Microbiome health during pregnancy is important for the successful birth of a healthy newborn infant. During the 3rd trimester of pregnancy, it has been shown that microbiome diversity and butyrate-producing *Fecalibacterium* decrease, and proinflammatory Proteobacteria and Actinobacteria spp. increase [10]. Complications of pregnancy related to abnormal changes in a pregnant woman’s microbiome include preeclampsia [11] and gestational diabetes mellitus [12]. Being overweight or being frankly obese negatively affects the intestinal microbiome during pregnancy [13]. A high-fat diet during pregnancy was shown to reduce the proportion of *Bacteroides* spp., an important constituent of the healthy infant microbiome early in life [14].

The vaginal microbiome changes during pregnancy. The major taxa identified in vaginal secretions before pregnancy are multiple species of the single genus *Lactobacillus* [15]. At the time of delivery, the vaginal microbiome is more complex and includes species from the phyla, Firmicutes, Proteobacteria, Bacteroidetes and Actinomycetales [16] as well as *Lactobacillus* spp. [17,18]. The vaginal microbiome in preterm birth primarily seen in women of African ancestry showed lower levels of *Lactobacillus* spp. and higher levels of bacterial vaginosis-associated bacterium 1 (BVAB1), *Sneathia amnii*, *Prevotella* spp. and a variety of other species [19]. 

### Antibiotic Use during Pregnancy

The frequency of antibiotic use in pregnant women has steadily risen on a global basis in recent years [20], with 20–40% given antibiotics for a variety of reasons, from prophylaxis to treatment of documented or suspected infection [21,22]. In a large study of pregnancies on the Danish registry, exposure to antibiotics during pregnancy increased risk of childhood infection-related hospitalizations [23]. Use of antibacterial drugs during pregnancy should be driven not by empiric use but by strong culturable evidence of presence of a treatable pathogen (e.g., group B *Streptococcus*).

Antibiotics not only significantly alter the gut and vaginal microbiome of the mother but, more importantly, also reduce the α-diversity (mean bacterial species density) in the gut and vagina of pregnant women, leading to a sustained drop in abundance of the most important taxa in newborns, *Bifidobacterium* and *Bacteroidetes* [24].

## 4. Important Elements in the Formation of the Microbiome and the Immune System of a Newborn

The first important microbiologic exposure of a newborn is at the time of delivery, with microbial growth in and on the infant being facilitated by immunological tolerance [25]. 

### 4.1. Natural Birth, Vaginal Delivery

The route of delivery has a major and persistent effects on the composition of the intestinal microbiota early in life [26]. In a study of the intestinal microbiome of infants at six months of age, those delivered vaginally had healthier and more diverse bifidogenic microbiota (Table 1) than infants delivered via cesarean section, who were shown to suffer from a greater number of respiratory infections [27]. 

Studies in a mouse model also found that vaginal delivery was more important in early shaping of the infant microbiome than exposure to the environment [28]. After birth, the timing of bacterial colonization of the gut while essential for health and development was shown to be time-dependent and variable for each infant [29]. 

While it is logical to assume that microbiota from the vagina are the critical early strains that help form a newborn’s microbiome during vaginal delivery, there are other pathways via which maternal microbiota reach the newborn, skin, tongue, and feces. There is growing evidence that taxa from the mother’s rectum and fecally-contaminated perineal surface are a more important sources of engrafting bacteria of the intestine of newborns during vaginal birth than the vagina [30,31]. In a supportive study, maternal vaginal organisms given orally to their respective infants born via cesarean section showed no modification of the infants’ intestinal microbiome compared with a placebo group [32]. In a second study, a pregnant woman with recurrent *Clostridioides difficile* infection (CDI) treated with fecal microbiota transplantation (FMT) lead to engraftment of FMT donor microbiota in the mother that were later transferred to her infant during vaginal delivery [33]. 

**Table 1 antibiotics-12-01617-t001:** Vaginal and Intestinal Microbiome Changes in Pregnancy and Intestinal Microbiome of Newborns.

Comparative Group	Microbiota Findings during Pregnancy	Reference
Pregnancy, Vaginal Microbiome	The vaginal microbiome influences pregnancy outcomes. During pregnancy, the vaginal microbiome was shown to undergo a reduction in α-diversity with reduced numbers of vagitypes of lactobacilli and lower prevalence of *G. vaginalis* and other microbiome profiles. Post-partum, the *Lactobacillus* spp. changed type and became less prevalent, and strains of Clostridia, Bacteroidia and Actinomycetia classes increased. Women who delivered preterm infants showed lower vaginal levels of *Lactobacillus* spp., higher levels of bacterial vaginosis-associated bacterium 1 (BVAB1), *Sneathia amnii* and *Prevotella* spp. and increases in proinflammatory cytokines in vaginal fluid.	[18,19,34,35]
Maternal Intestinal Microbiome During Pregnancy	Microbiota diversity was shown to decrease in the 3rd trimester with an increase in strains of Proteobacteria and Actinobacteria and decrease in strains of butyrate-producing, anti-inflammatory *Fecalibacterium*.	[10]
Intestinal Microbiome During Pregnancy with Obesity	Dysbiosis was seen with an increase in strains of Firmicutes and an increased in *Staphylococcus* and Enterobacteriaceae Decrease in *Bifidobacterium* and *Bacteroides*.	[36,37]
Pregnant Women Given Antibiotics	Antibiotics significantly altered the gut and vaginal microbiome but more importantly reduced the α-diversity in the gut and vagina of pregnant women, leading to a sustained drop in abundance of *Bifidobacterium* and Bacteroidetes in newborn.	[38]
	Infant Microbiome Findings	
Vaginal Birth	The most abundant phyla in the first-pass meconium, reflecting the in utero transfer of microbes to a newborn, were shown to be strains within the phyla, Firmicutes, Proteobacteria and Bacteroidetes. During the first few days of life, the intestinal microbiome of newborns delivered vaginally showed high proportions of *Bacteroides*, including *Prevotella* spp., *Bifidobacterium* and *Lactobacillus*.	[16,39]
Cesarean Delivery	In infants born via cesarean delivery, α-diversity of fecal microbiota was reduced with colonization by environmental bacteria, *Staphylococcus* spp. and *Propionibacterium*, without vaginal organisms such as *Lactobacillus*, *Bacteroides* (including *Prevotella* spp.), and *Bifidobacterium* spp., which were only found later. Cesarean delivery also influenced infant microbiome with breast feeding early on when compared with vaginal delivery, which may have implications for infant health.	[26,40,41]
Perinatal Exposure to Antibiotics	Perinatal exposure to antibiotics led to reduced fecal α-diversity and reduced maturity of the microbiome with depletion of health-promoting *Bifidobacterium* spp., *Lactobacillus* spp. Lachnospiraceae and Bacteroidetes spp. and with increase in proportion of proinflammatory Proteobacteria. Some species were found to be dominated by a single strain, and antibiotic resistance genes carried on microbial chromosomes decreased sharply, while those on mobile elements were shown to persist long after antibiotic therapy was stopped.	[39,42,43]
Preterm Birth	Preterm infants show reduced microbiome diversity and colonization by facultative anerobic bacteria, *Enterobacter*, *Enterococcus*, *Escherichia* and *Klebsiella* and showed a higher frequency of antibiotic resistance if antibiotics were administered. Factors that negatively influenced microbiome development with preterm births included prolonged hospitalization, postnatal medications and formula feeding.	[44,45,46,47,48]
Breast feed infants	Breast feeding was shown in one study to be the major factor determining the composition of infant microbiomes. The most abundant bacterial species in exclusively breastfed infants were *Bifidobacterium* spp., *Lactobacillus* spp. and *Bacteroides* spp. with lower fecal α-diversity compared with formula fed infants.	[36,49,50]
Formula Fed Infants	Despite the efforts of manufacturers to make formula preparations as close to breast milk as possible, the microbiome findings show there remain differences. Common strains found in formula-fed infants were members of *Streptococcus*, *Enterococcus*, Lachnospiraceae, Enterobacteriaceae, Staphyloccoccaceae and *C. difficile* with low levels of *Bifidobacterium.* Antibiotic-resistance genes were found to be present in one study.	[49,51]
Changes After Weaning	As infants begin solid foods after weaning from breast feeding or infant formula, they are exposed to dietary carbohydrates (glycans) that alter the gut microbiome. The introduction of solid foods alters the gut microbiota moving toward a more mature microbiome seen in older children, with the most prevalent strains from the phyla Firmicutes and Bacteroidetes and with an increase in *Atopobium*, *Clostridium*, *Akkermansia*, Lachnospiraceae and *Ruminococcus* and a decrease in facultative anaerobes.	[52,53,54]
Changes when attending a Day Care Center	While home care infants and young children commonly were found to be colonized by *Bifidobacterium* spp. and *Lactobacillus* spp., infants attending DCCs had microbiomes more closely resembling those of older children in age-match studies with increased abundance of species within the Firmicute phylum (including species of Lachnospiraceae and *Ruminocccus*), and *Bacteroides* (including *Prevotella* spp).	[55]

### 4.2. Cesarean Delivery

In the US, the C-section rate remained at 22% from 2016 to 2019 and then rose an additional 4% from 2019 to 2021 [56]. In a study of 9350 deliveries carried out in 2001, 11.6% underwent a non-medically indicated cesarean delivery, providing indirect evidence that many cesarean deliveries are medically unnecessary [57].

Cesarean delivery excludes the newborn from vaginal and pelvic microbiota, leading to acquisition of a less diverse intestinal microbiome [58], which may explain why the intestinal microbiome of newborns born via C-section less resemble the gut microbiome of their mothers when compared with infants born vaginally [59]. In one study, microbiota acquired during cesarean delivery were acquired from the general environment in the hospital [60]. In a systematic review of infants, the gut microbiota for the first 3 months of life were affected by mode of delivery [61], with environmentally acquired organisms common constituents of the intestinal microbiome in newborns born via cesarean delivery [62]. The microbiomes of babies born via cesarean delivery begin to resemble breast-fed babies between 4 and 12 months of age [59], but by then, the health benefits exerted by a diverse microbiome early in life have been lost. Babies born via cesarean delivery suffer more frequently from immune-mediated and allergic disorders, including asthma [63,64,65], inflammatory bowel disease, celiac disease, obesity [64] and type 1 diabetes [66] than those born by vaginal delivery.

The practice of giving mothers undergoing C-sections antibiotics to reduce infections further impairs the microbiome of both the mother and newborn. In one study of cesarean delivery, the findings of an increased ratio of Proteobacteria/Bacteroidetes and fecal colonization of *C. difficile* at 12 months after cesarean delivery were markers of unhealthy microbiome and a predictor of later childhood obesity and atopy [67].

### 4.3. The Role of Host Genetics in Shaping the Infant Microbiome

A study of monozygotic and dizygotic twins was carried out to determine the relative importance of genetics versus environmental factors in the formation of the early microbiome [68]. At month one, the monozygotic pair showed common patterns in their fecal microbiomes distinctly different from those of a fraternal sibling, but by one year of age they showed a similar microbiome pattern. The authors hypothesize that genetic factors are important in early infant life. Abundance of two core organisms acquired early in life, *Bifidobacterium* and *Ruminococcus* [69], was shown to be dependent on the presence of two human genes [70]. Genetic factors are less important in influencing constituents of the microbiome of older children and adults [71,72].

### 4.4. Formation of the Infant Immune System

The immune system undergoes a programming and maturation process early in life [2], facilitated by the immunological tolerance of newborns that occurs as a response to regulatory T lymphocytes from their mothers [62], allowing colonization by organisms encountered early in life. In experimental studies in a germ-free mouse model, the exposure to microbes was essential to the development of a complete functioning immune system, but the exposure had to occur at an early age [73]. The microbiome participates in both pro-inflammatory and regulatory responses of the immune system [74]. Most young children’s microbiome diversity pattern matures into that seen in older children and adults after 2 to 3 years of age [75], or it may take up to 5 years of age or even longer [76]. The infant’s microbiome begins to participate in the regulation of the immune function via interphase with intestinal intraepithelial lymphocytes and the gut immune system [77] leading to IgA-coating of a proportion of gut bacteria, first from breast milk [78] and then, after a few weeks, from the infants’ own immune system [79], providing a homeostatic effect in a setting of microbiome health, or in the setting of dysbiosis, coating of proinflammatory bacteria in attempting to attenuate microbial virulence [80]. 

Engraftment of a healthy diverse microbiota is important to development of early infant health, and its absence, together with reduced microbiome diversity (dysbiosis), can lead to alteration of the immune system and development of atopic disorders and food allergies [81,82]. Infant immune training and maturation during early life can prevent later immunologic disorders [83]. Studies in pathogen-free vs. germ-free mice have demonstrated that age-specific exposure to microbes during childhood is associated with protection from immune associated disorders via reduction of natural killer T cells that, when present, predispose those affected to asthma and inflammatory bowel disease. A healthy microbiome is important in immune response to an infecting organism, to response to vaccines [3] and to response to cancer chemotherapy in children [84].

### 4.5. Breast Feeding and Infant Diet in Microbiome Development

Breast feeding has become accepted by most pregnant women in Western cultures. In a very large survey of different ethnic groups in the United States, 88% of mothers delivering infants decided to start breast feeding [85]. Reasons not to start included “didn’t want to”, “didn’t like it” or “taking care of other kids”. At 10 weeks, 70% were still breast feeding. The mothers who stopped indicated they had “trouble with baby latching”, “breast milk was insufficient” or they had “nipple pain”. 

There are multiple reasons why future mothers should be encouraged to breast feed their newborn infants. A study of 107 mother infant pairs found that bacteria in the mother’s milk engrafted in the colon of their infants became an important source of the infants’ microbiome [86]. Breast milk is a complex biofluid that contains a number of active ingredients that exert immunomodulatory effects [87], such as hormones [88], with perhaps the principal effect on gut microbiome exerted by the different indigestible oligosaccharides contained in human milk [89]. Finally, breast milk provides excellent nutrition to infants. Breast feeding brings macronutrients and micronutrients and other well-defined factors to the infant, which contribute to a newborn’s microbiome formation, which is additionally influenced by the mother’s diet [90,91]. The diet of an overweight mother can affect her infant’s weight, causing metabolic dysfunction, and is one of several factors that can lead to obesity and type 2 diabetes in infants [92]. Undernutrition in childhood leading to dysbiosis was shown in one study to increase risk of later development of coronary artery disease [93].

Complementary food can be added to the infants’ diet at about 4 months of age, when the microbiome of the infant shows major differences from the mother’s microbiome [59], contributing to the expansion and complexity of the infant’s microbiome. Cessation of breast feeding sometime around six months of age was shown to correlate more with the establishment of an adult-type microbiome than introduction of solid food in a Swedish study [59]. It is particularly important to delay feeding from the tabletop in the rural developing world to prevent enteric infection from contaminated foods [94].

### 4.6. Hygiene and the Environmental

While physical interaction between mother and infant contributes the greatest to the early development of infant microbiomes, direct exposure to microbiota because of hygienic factors or unique environments, such as exposure to soil, animals and other people all contribute to the evolution of the microbiome and the general health of young children [95].

A setting where environmental contamination may help establish microbiome diversity with programming of the immune system leading to improved health during later years is on farms with exposure to animals. Children growing up on farms show reduced frequency of later asthma [96], other allergies and inflammatory bowel disease [97,98]. In a study of 82 mothers, cord blood obtained from the 22 farming mothers showed higher levels of T regulatory cells and lower cytokine levels and lymphocyte proliferation than non-farming mothers, indicating differences in immune development in the two study groups [99]. In another study, early exposure to a microbial world seen with farming and exposure to cats and dogs in infants was shown to lead to development of an IFN-γ immune responses during the first 3 months of life [100]. 

Day care centers (DCCs) housing young children, typically beginning at about 3 months of age, put infants together with other non-toilet-trained infants, contributing to fecal contamination of the environment [101]. In the early weeks of first attending large day care centers, children often experience increased rates of upper respiratory infections and bouts of infectious diarrhea which lesson in frequency with continued presence in the facilities [102,103]. Unique microbiome engraftment was seen in children attending one of four DCCs compared with age-matched children living at home [55]. The DCC effect on a child’s microbiome may relate to size of the center and exposure to areas in the DCCs where there is a greater likelihood of fecal contamination [104].

In one study, the frequency of attending day care centers or having close interactions with other children during early years of life was shown to inversely relate to the frequency of diabetes [105,106]. In large families, young siblings were shown to have reduced rates of atopic disorders compared to young children from smaller families [107,108]. 

Early microbial exposures from the environment have been postulated to train the immune system not to overreact to immune stimuli [55]. The “hygiene hypothesis” that focuses on the importance of exposure to a microbial world to improve health should not be abandoned [109], as it represents important pathways for microbiome development in infancy and childhood. Two important environmental sources of microbiota in early life that contribute to the diversity of the microbiome are exposure to dirt and other children, who lack hygienic standards. Soil and the human intestinal microbiome were shown to contain similar concentration of microbiota and specific taxa [110].

A rich and diverse microbiome was shown to be present in African hunter–gatherers with limited hygienic practices, who have near-constant exposure to environmental microbes from the ground, animals and people and rarely receive antibiotics [111]. From this close-to-earth population, the research team found a richer microbiome with more than two times the number of bacterial species than in their U.S. control population and with organisms shown to be less prone to oxidative damage. 

### 4.7. Perinatal Antibiotics

It has been estimated that between 2% and 5% of newborns are exposed to parenteral antibiotics for presumed sepsis [112]. A prospective controlled study of 100 term newborns delivered via the vaginal route were studied for exposure to pre-natal and post-partum antibiotics to determine the effects on the fecal microbiota at one year of age in study conducted in Finland [113]. Perinatal antibiotics severely damaged the intestinal microbiome, which persisted for at least until one year of age, a critical duration of time for immune system development. Microbiome damage was shown to be far greater in infants than when antibiotics were given to older children. Dysbiosis from early infant exposure to antibiotics in other studies found the impaired microbiome was still present at 2–3 years of life when studied, a critical time for microbiome health [39,114]. Persistent damage to the microbiome in young children can result in future allergic and metabolic consequences [115], including asthma, atopic disorders, obesity, type 1 diabetes and inflammatory bowel disease [116]. Additionally, prior antibiotic exposure in early life can encourage development of antibiotic resistant gene reservoirs in their microbiome that make them more susceptible to difficult-to-treat infections [117]. 

With the National Ambulatory Medical Care Survey 2010–2011, it was found that 30% of outpatient antibiotic prescriptions were inappropriate, with the largest number being directed to children with respiratory infections not meeting standard criteria for treatment [118]. An effective national and local antibiotic stewardship program should be developed to minimize inappropriate use of antimicrobials targeting infants [119]. Greater efforts to document infection via laboratory testing rather than giving empiric treatment and using the narrowest spectrum antibiotics when this treatment is needed should be followed.

## 5. Microbiome Patterns Seen in Mother Infant Settings

Table 1 shows the expected pattern of bacterial taxa seen in various maternal–infant settings and exposures. The microbiome patterns discussed are found in the vagina and intestine during pregnancy, when pregnant women receive antibiotics, and in infants by modes of delivery, preterm birth, breast- vs. formula-fed and during the weaning process. 

## 6. Current Microbiota Therapy to Improve Microbiome Diversity in Pregnancy and Infancy

Figure 1 outlines the approaches for microbiota therapy for pregnant women and newborn infants to improve their microbiome health. On the left side of the figure, use of prebiotics and probiotics are outlined, which are available for use by patients or physicians in improving the microbiome in pregnancy and infancy. Prebiotics (soluble fiber) select for the growth of bacteria that produce metabolites such as short chain fatty acids that contribute to a healthy microbiome. Probiotics, defined as living organisms with health benefits, are designed to engraft in the colon and add to the diversity of the microbiome. On the right side of the figure are ways to improve the diversity of the microbiome in research centers where support is needed to obtain approval from the federal and local ethical committee. Vaginal seeding and fecal microbiota transplantation for infants delivered via cesarean delivery, especially if breast feeding is not planned, can produce a healthier and more diverse microbiome with microbiota important to the formation of the immune system. New and novel products are in development that show intestinal engraftment and provide health benefits. Each of these will be described in detail. 

### 6.1. Prebiotics

Common prebiotics that should be part of all diets include non-digestible fibers or resistant starches that reach the colon and stimulate growth of bacteria releasing beneficial metabolites such as short chain fatty acids. Breast milk contains more than 200 different types of complex carbohydrates classified as oligosaccharides, which reach the colon intact and stimulate growth of *Bifidobacterium* spp., *Bacteroides* spp. and other healthy infant bacteria that protect the epithelial lining, prevent abnormal gut wall permeability, have antibacterial properties and facilitate infant gut immune system programming [120]. Milk formula companies have tried to match the oligosaccharides in breast milk with synthetic or animal milk-derived oligosaccharides, galactooligosaccharides (GOSs), fructooligosaccharides (FOSs) and polydextrose, which reduce rates of occurrence of infant colic [121] and modify the microbiomes of infants to be more like those seen in breastfed infants than those fed previous versions of infant formulas [122]. There is currently a lack of studies showing that these fortified infant formulas have the same health benefits as breast milk. 

### 6.2. Probiotics

As seen in Table 2, there are positive and negative studies using probiotics to improve maternal–infant health by increasing the diversity of the microbiome both in pregnant women and in newborns.

Probiotics have commonly been used to improve dysbiosis seen in infants delivered via cesarean section and to treat and prevent metabolic alterations associated with pregnancy, including weight gain, gestational diabetes and changes in lipid profile. Additional benefits sought from probiotic use in mothers and their infants were correction of dysbiosis, prevention of gestational diabetes, prolongation of gestational age of newborns and prevention of future medical disorders in the infant, most importantly allergies, atopic dermatitis and metabolic complications. Colonizing the infant with healthy bacteria after cesarean delivery has been tried using probiotics [123]. As seen in Table 2, there are both positive and negative results from clinical trials examining use of probiotics during pregnancy and in infancy. 

**Table 2 antibiotics-12-01617-t002:** Research Approaches to Improve Maternal–Child Health via Microbiota Replacement Using Prebiotics or Probiotics (* If not specified, most probiotic studies employed one or more strains of *Lactobacillus* and *Bifidobacterium*).

Evaluation of Microbial Products to Improve the Microbiome and Health		
Pregnant Women		
Study Outline	Probiotics Used	References
**Positive Studies**		
Treatment improved diversity of intestinal or vaginal microbiome and insulin sensitivity, and reduced inflammation.	VSL#3 (Visbiome) (8 strains La, Lp Lc Ld Bb, Bl and Bi)	[124,125,126]
Probiotics in pregnant women partially protected infants from atopic dermatitis mediated via the reduction of Th22 cells.	LrGG, Ba, and La, LrGG and Bl	[127]
Probiotics given to pregnant women with GDM and/or obesity in controlled clinical studies decreased fasting glucose levels, increased insulin sensitivity and improved lipid metabolism compared to placebo treatment.	VSL#3 (8 strains La, Lp Lc Ld Bb, Bl and Bi)	[128,129,130,131]
During last trimester, increased microbiota diversity in vagina and reduced anti-inflammatory cytokines	VSL#3 (8 strains La, Lp Lc Ld Bb, Bl and Bi)	[126]
**Negative Studies**		
Metabolic value of probiotics was not found in three clinical trials during pregnancy.	LrGG, Ba, Ls	[132,133,134]
**Pregnant Women and Offspring**		
**Positive Studies**		
A cohort of 159 overweight or obese pregnant women were given a probiotic or placebo four weeks before expected delivery, and the dose was continued in infants for 6 months postnatally. Perinatal probiotics moderated early and later weight gain in the infants.	LrGG, Bl 2	[131]
Prevention of atopic dermatitis occurred in infants given probiotics if their baseline microbiome was diverse, without dysbiosis.	A combination of strains of LrGG, Bb and Pf	[135]
When probiotics were given to pregnant women with an unborn fetus at risk for allergic disease and then given to the newborn allergic disorders were prevented during 13 years follow-up only in infants undergoing cesarean delivery.	LrGG, Bb, Pf	[136]
A cohort of 27 infants (mean age 4.6 months) who experienced atopic eczema during exclusive breast feeding responded clinically to probiotics with weaning in a controlled study.	Bl, LrGG	[137]
Administering probiotics to post-partum women lead to presence in breast milk of beneficial bacteria that controlled infant weight and reduced occurrence of infant colic in randomized controlled clinical trials.	Multiple combinations of probiotics were used *	[138]
Studies were reviewed where probiotics were given to pregnant or post-partum women to see the effect on their infants. Durable reduction in atopic eczema for up to 7 years in the infant was seen in one study; other studies either showed mixed results in reducing eczema; additional findings of the studies included reduced blood glucose and increased glucose tolerance during pregnancy, and immunologic findings suggested a positive effect including increasing the amount of anti-inflammatory cytokine transforming growth factor β2 in lactating mother’s milk, likely with immunoprotective effects.	LrGG, Lr	[139]
Preclinical and clinical studies of the biologic effects of probiotic use in pregnancy were reviewed. Evidence was summarized to show that selected probiotics had programming potential for sustained benefit to offspring. Effects of probiotics on the infants included improved growth indices, intestinal barrier function, neurodevelopment, resistance to allergic disorders and metabolic disease, and increased diversity of intestinal microbiota.	Multiple combinations of probiotics were used *	[140]
A total of 28 randomized controlled clinical trials involving 4865 study participants from 2010 to 2020 were selected for meta-analysis. The analyses showed probiotic supplementation had an effect in decreasing GD-predisposing metabolic markers such as blood glucose, lipids, inflammation and oxidation, which may reflect an effect on reduction of GD in pregnant women.	Multiple combinations of probiotics were used *	[17]
In a meta-analysis of probiotic supplementation in pregnant women for prevention of GDM, 10 randomized controlled trials were included. A correlation was found between probiotic use and fasting serum insulin and insulin resistance. No significant correlation was seen between probiotic use and lipid levels in pregnant women with GDH. For healthy pregnant women, probiotics were negatively associated with fasting serum insulin. No correlations were found between probiotic use in fasting plasma glucose.	Multiple combinations of probiotics were used *	[141]
In a meta-analysis looking at improved glucose and lipid metabolism in pregnant women, 10 randomized clinical trials were reviewed. Probiotic use in this study led to a reduction in fasting blood glucose, serum insulin levels and insulin resistance in early pregnancy felt to represent positive effects in reducing the risk of GDM.	Multiple combinations of probiotics were used *	[142]
**Negative Studies**		
Use of prebiotics in pregnant women with obesity was studied in six human trials and four animal studies. The research failed to show a positive impact in metabolic health in the women or their offspring.	Multiple combinations of probiotics were used *	[143,144]
Probiotics given to 31 pairs of healthy pregnant women and newborns did not lead to improved diversity of the microbiomes compared with controls, although there were differences in microbiome communities or networks.	BL, LDB, St	[145]
Randomized controlled trials were examined via the Cochrane Pregnancy and Childbirth’s Trials Registry dealing with probiotics in prevention of gestational diabetes mellitus looking at both the mother and the infant. Overall, the certainty of evidence for a probiotic effect appeared to be low in pregnancy and during early childhood. The study expressed uncertainty about a number of probiotic effects versus placebo in 9 studies looking at GDM, in 3 studies looking at for effects on blood pressure, gestational age of infants in cesarean section (3 studies), for a difference in induction of labor (1 study), occurrence of heavy bleeding immediately after birth, weight gain during pregnancy or total gestational weight gain, or difference in fasting blood sugar (7 studies). There was a slight reduction in triglycerides and total cholesterol (4 studies) and a reduction in insulin secretion with probiotics (7 studies). The study expressed uncertainty about effect on newborns birthweight, gestational age at birth, preterm births, large weight babies or need for admission to intensive care units.	Multiple combinations of probiotics were used *	[146]
**Newborns and Young Infants Studies**	**Prebiotics Given**	
Breast milk prebiotics to fortify the infant formulas with galactooligosaccharides (GOSs) and fructooligosaccharides (FOS) that were shown to reduce infant colic and improve the infant gut microbiome.	GOS and FOS	[120]
Caution has been put forth to indicate that probiotics administered to full-term infants to improve dysbiosis may have adverse events if used in low-birth rate infants.	Bifidobacterium sepsis occurred in low-birthweight infant and in another neonate with omphalocele	[147,148,149]

GDM = gestational diabetes mellitus, GD = Gestational diabetes, TG = triglycerides, VLDL cholesterol = very low-density lipoprotein cholesterol. VSL#3 is a probiotic combination licensed in 40 countries; it is also called Visbiome and contains 8 different probiotic strains listed in the table. *Lactobacillus* probiotic strains: *L. acidophilus* (La), *L. plantarum* (Lp), *L. casei* (Lc), *L. delbrueckii* subspecies *bulgaricus* (Ld), *L. reuteri* (Lr), *L. rhamnosus GG* (LrGG), *L. salivarius* (Ls). *Bifidobacterium* probiotic strains: *B. breve* (Bb), *B. longum* (B.l 1), *B. lactis* (Bl 2), *B. infantis* (Bi), *B. animalis* subsp. *Lactis* bb-12 (Ba). *Propionibacterium freudenreichii* (Pf). *Streptococcus thermophilus* (St).

The many controlled clinical trials evaluating currently available probiotics in pregnancy and early infancy that are briefly outlined above have shown mixed results. Most of the currently available probiotics employ a mixture of strains of Lactobacillus spp. and Bifidobacterium spp., two genera with a long history of safety [150] that meet criteria for production, good growth and tolerance for the stresses of preparation and storage [151]. To be commercially successful, the probiotic strains need to be stable at ambient temperature and humidity for at least 24 months [151].

### 6.3. Vaginal Seeding

Research has shown a physiologic benefit to infants delivered via cesarean section, namely that of exposure to vaginal microbiota obtained from their mothers before delivery, after screening them for pathogenic microbes (e.g., group B streptococci, *C. trachomatis*, *N gonorrhea* and human papilloma virus) [152]. The microbiota seeding approaches taken have varied from placing a sterile gauze pad soaked in saline inserted in the mother’s vagina for one hour before performing cesarean delivery [153,154], to using multiple vaginal swabs obtained from pregnant women just before delivery, stored and applied to newborns’ lips, then to their entire bodies [155]. In a study of 68 pregnancy women, administration of vaginal swabs applied to their respective cesarean-delivered infants, in a placebo-controlled study, found that the gut microbiome and metabolome were improved in the actively treated infants with improved neurodevelopment at six months [155]. 

Vaginal seeding after a cesarean delivery remains a research approach and should not be considered standard of care. Standards for screening of the mother have not been developed, the few trials so far conducted have been small, and treated subjects have not been followed long enough to be certain of longer-term effects. The American College of Obstetricians and Gynecologists has indicated vaginal seeding should only be performed using an IRB-approved research protocol until the method has been standardized and approved by an appropriate consensus development committees [152].

### 6.4. Fecal Microbiota Transplantation (FMT)

Since it appears likely the important bacteria acquired by infants delivered vaginally have their origin from the maternal fecal pool, one approach to populate the infant gut with healthy bacteria after cesarian delivery has been via FMT using the mother as donor, where a fecal suspension is mixed with the mother’s breast milk and administered orally to the newborn [156]. This approach was shown to provide infants born via cesarean delivery with microbiomes like those seen in children born via vaginal birth. 

Instructions for performing FMT in a newborn infant have been published [157]. The maternal safety screening would need to be updated with a national organization such as the U.S. FDA, which monitors FMT donor screening to provide the greatest safety assurance.

### 6.5. FMT in Pregnancy for Recurrent CDI

FMT has been used as single treatment in pregnant women with recurrent CDI with success [33,158]. In one of the studies where FMT was given to a pregnant woman for recurrent CDI, bacterial strains engrafting the pregnant woman also engrafted her infant’s gastrointestinal tract [33].

### 6.6. FMT Is Not Needed in Infants for the Diagnosis of CDI

Infants are frequently colonized with *C. difficile* at birth and appear to be resistant to infection and clinical illness from the organism. It is difficult to make a diagnosis of CDI in infants because so many have positive toxin tests normally. An expert group concluded that CDI in infants is rare if it exists at all [159]. A policy statement from the American Academy of Pediatrics, published in 2013, stated testing of infants younger than 12 months of age for *C. difficile* is complicated by the high rate of asymptomatic colonization, and alternative etiologies (for diarrhea) should be sought even in those with a positive test for *C. difficile* [160]. 

As cases of CDI in infants appear to be rare [161], CDI recurrences and need for fecal microbiota transplantation (FMT) for this infection do not exist for infants. 

## 7. New Advances in Microbiota Therapy in Development

With the knowledge that the vaginal and intestinal microbiome of a pregnant woman and the intestinal microbiome of her offspring are important to the current and future health of the infant, new approaches to microbiota replacement strategies are in development. 

### 7.1. Prebiotics

Soluble and insoluble fiber and resistant starches will improve the diversity of the gut microbiota of pregnant women [2,162], which can reduce allergic symptoms in newborns after delivery [163]. Other prebiotics being administered are fructans (FOS and inulin), GOS and polydextrose, which improve the diversity of microbiomes in pregnant women and in infants via their formulas [164,165]. 

### 7.2. Novel Probiotics

While the currently available probiotics were developed before the Human Microbiome Initiative and contemporary understanding of the biology of the microbiome, we now have the methods to develop modern microbiota replacement probiotics and biologic agents. *Lactobacillus* reuteri, strain DSM 17938, has been rediscovered as a bioactive probiotic that colonizes the intestine and shows immune responsiveness and may well be an effective treatment of necrotizing enterocolitis in premature infants [166]. 

Another approach is to use bacterial strains important in a healthy human microbiome as a probiotic cocktail, including strains of *Bacteroides*, *Clostridium*, *Roseburia*, *Alistipes*, *Fecalibacterium*, *Prevotella*, *Blautia* and *Akkermansia* spp. [167]. It is hoped that by combining strains with known physiologic roles in the human microbiome, synergistic or additive effects will be seen. 

Molecular modification of enteric bacteria or yeast is another approach to probiotic development. Genetic engineering of probiotics may result in improved delivery of bioactive molecules that inhibit pathogenicity or focus on regulatory systems or molecular targets important in disease or improved health by altering a physiologic pathway such as inflammation, rather than providing broad antimicrobial effects. Such modifications have included recombinant strains that have anti-inflammatory, hormone-secreting, chemical-reducing functions targeting a specific disease [168,169,170]. Gene editing may be facilitated via available tools such as CRISPR-Cas technology [171,172]. Engineering probiotic strains for specific biologic activity via enhanced gene expression can lead to a product with desired properties for an indication [173]. 

An additional approach to favorably modify an abnormal microbiome is to harness bacteriophages, DNA viruses which have the capacity to infect and control essentially all intestinal microbiota, thus fine-tuning the microbiome for functional control [174].

### 7.3. Synbiotics

An important research line to pursue in designing microbiota therapy is development and testing of synbiotics, the combination of live microorganisms (probiotics) and selective growth-promoting substrates utilized by the microorganisms (prebiotics) that confer health benefit to the host [175]. 

## 8. Conclusions and Future Directions

In contrast to microbiome resilience and stability in older children and adults on a regular diet without receiving antibiotics, the gut microbiome is under rapid evolution in newborns and young infants and sets the stage for future health. The first 100 days of life represent the most important time for the human microbiome, where associated physiologic events include formation of the immune system, and the establishment of body metabolism network and brain maturation occur.

Earlier studies before our current understanding of microbiome health identified an important role of intrauterine and infant health in determining health later in life. Barker et al. [176] performed epidemiologic studies of 8760 subjects in England, showing that low birth weight and low BMI at 2 years of age were associated with later insulin resistance and coronary artery disease. What evolved from Barker’s observations and others was a broader view of the importance of health in infancy, called the Developmental Origins of Health and Disease (DOHaD), which focused on the association between birth weight and disease outcomes later in life [177]. The breakthrough which has led to our modern concepts of maternal–infant health and the role of the microbiome in mediating the key events was the NIH Human Microbiome Project initiated in 2007 [178] and the Integrative Human Microbiome Project [179]. 

The most obvious approach to optimizing the microbiome of a pregnant woman is a healthy diet via consumption of prebiotics [180]. Nutritionists should become a regular part of care of women during pregnancy. Additionally, antibiotics should be given during pregnancy only with culture evidence of a treatable pathogen with approval of the women’s obstetricians. 

In pregnant women who have received courses of antibiotics, suffer from obesity or have known dysbiosis from another condition, commercially available probiotics may be considered and, if used, monitored by the patients’ obstetricians. It is difficult to recommend a specific probiotic since there have been essentially no comparative trials. Not being controlled as drugs, standardization for production is not guaranteed. The U.S. FDA inspected 656 facilities producing dietary supplements in 2017 and found significant violations in more than half of the facilities, such as failure to establish the purity, strength or composition of the product [181]. There is need to develop more active probiotic strains with important biologic activity than currently available. The ideal product will be in pure form and safe to administer. 

Traditional approaches to maternal–infant health should be pursued when feasible, including the maintenance of good health of the pregnant woman and encouraging pregnant women to pursue both traditional vaginal delivery and breast feeding. Where these are not possible, microbiologic approaches may be considered to improve diversity of the microbiome in attempting to improve health of newborn children. In research centers, for patients who have been delivered via cesarean delivery, vaginal seeding or fecal microbiota transplantation can be considered as part of an IRB approved clinical trial. Also, in research centers, FMT can be considered for use to treat CDI in pregnant women depending upon severity of symptoms. Purified bacterial strains can be used to reverse dysbiosis in pregnant women and infants with dysbiosis. 

For future directions, developing purified microbial products to replace probiotic mixtures and fecal-derived products to reverse the microbiome will improve product safety and consistency. The first licensed product that meets these criteria in the U.S. is VOWST (investigational drug SER-109), live purified spores of non-*C. difficile* Firmicutes [182] that, when administered orally, successfully cures recurrent CDI [183]. VOWST has not yet been evaluated in pregnancy or infancy.

Limitations of the review included inability to review all papers dealing with the subject, the large variability in quality of studies reviewed and discussed and gaps in the literature regarding important science needed to fully understand the topic. Also, the probiotic studies performed were small, and comparisons between probiotic compounds and combinations were not performed.

## Figures and Tables

**Figure 1 antibiotics-12-01617-f001:**
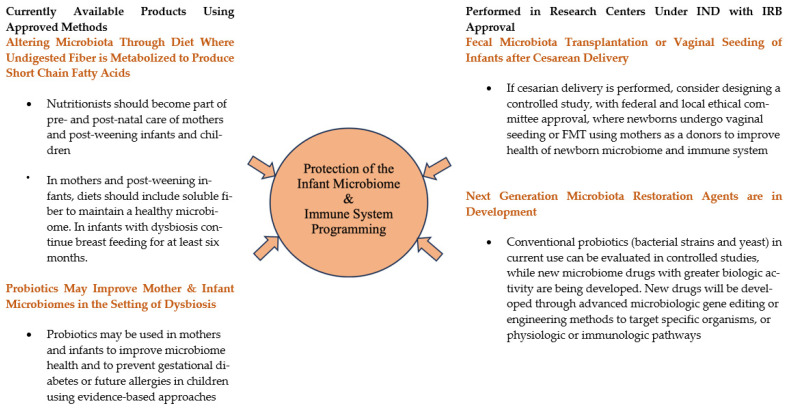
Microbiota Therapeutic Strategies to Improve Microbiome Health of Pregnant Women and Later Born Infants.

## Data Availability

Not applicable.

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
