# Peer review of "The Importance of a Healthy Microbiome in Pregnancy and Infancy and Microbiota Treatment to Reverse Dysbiosis for Improved Health"

_antibiotics, 2023, doi:10.3390/antibiotics12111617_

Round 1

Reviewer 1 Report

Comments and Suggestions for Authors

The authors have conducted an extensive review on Importance of a Healthy Microbiome in Pregnancy and Infancy.

The manuscript is very complete and well structured but information on the methodology and limitations of study is absent.

I suggest to include the description of the methods used to search for the articles. For example, keywords used, search languages, types of publications included, search period, inclusion, exclusion and elimination criteria of the articles, etc. Before the conclusions, a paragraph should be included about the limitations of the study.

Comments on the Quality of English Language

There are some spelling mistakes in pag 3, 4 and 13 (see attached file)

Author Response

Reviewer 1

The authors have conducted an extensive review on Importance of a Healthy Microbiome in Pregnancy and Infancy.

The manuscript is very complete and well structured but information on the methodology and limitations of study is absent.

I suggest to include the description of the methods used to search for the articles. For example, keywords used, search languages, types of publications included, search period, inclusion, exclusion and elimination criteria of the articles, etc. Before the conclusions, a paragraph should be included about the limitations of the study.

A methods section has been added just after the introduction and a limitation section near the end of the discussion. These were important suggestions.

Comments on the Quality of English Language

There are some spelling mistakes in pag 3, 4 and 13 (see attached file)

I didn’t see your file, but with complete revision of the paper, I hope now these are corrected.

Reviewer 2 Report

Comments and Suggestions for Authors

-Please illustrate the figure and the detail to describe the mechanism of any change to the composition of resident commensal communities comparing to the community found in healthy individuals.

-Please describe deep into the approaches for microbiota therapy to pregnant women and newborn infants to improve their microbiome health according to figure1.

-Please indicate the Table 2 in the main content of the manuscript.

-Please rearrangement for the order of references in Table1 accordingly.

-Please correct for the reference format.

Comments on the Quality of English Language

Minor editing of English language required.

Author Response

Reviewer 2

My comments are underlined. 

While I am not sure the meaning of each of these suggestions, I did my best.

-Please illustrate the figure and the detail to describe the mechanism of any change to the composition of resident commensal communities comparing to the community found in healthy individuals.

For the Figure for each of the items discussed I added a mechanism whereby each contributed to the microbiome.

-Please describe deep into the approaches for microbiota therapy to pregnant women and newborn infants to improve their microbiome health according to figure1.

I added additional description of each of the points discussed in the table in the first paragraph after Microbiota Therapy to Improve Microbiome Diversity in Pregnancy and Infancy

-Please indicate the Table 2 in the main content of the manuscript.

Table 2 has not been expanded.

-Please rearrangement for the order of references in Table1 accordingly.

The references in the Tables and Figure were arranged according to where the table was mentioned in the paper.

-Please correct for the reference format.

The format seems correct in the proofs sent to me.

Comments on the Quality of English Language

Minor editing of English language required.

I hope you are satisfied with the English now. English is my native language and I have gone over it carefully.

Reviewer 3 Report

Comments and Suggestions for Authors

Specific Comments

1.    Line 13: simplify the Background and Conclusion sections into one sentence, respectively (remove the term “background” and “conclusion”; expand and elaborate the Finding section (remove the term “Findings”)

2.    Employ another way to highlight your subheading, such as Line 77, 82, 130.

3. Subheadings from different levels should be distinct and easily recognizable.

4.    Only the “Introduction” (Line 31) and “The Microbiome during Pregnancy” (Line 47) are bolded. Maintain uniform formatting for all levels of headings.

5.    In Table 1, “Infant Microbiome Finding” should be bolded as well. Try to simplify the description using bullet points.

6.    Would it read smoother if position mother-infant comparison after Line 193 (when you have discussed both sides).

7.    Categorize Line 266 to 392 under “current microbiome treatments.”

8.    It would be helpful for the audience if there is another table/figure highlighting the disparities between previous and current/future probiotics treatment.

9.    Enhance the conciseness of Figure 1 by using visual aids and bullet points. Avoid citations in the figure.

10. Could you provide a brief description every time you introduce a new abbreviation? First example in table 2 under “Pregnant Women”, what does VSL#3 stand for? Also, Line 457, what is VOWST?

11. Line 289 “As seen in Table 2, …” could be added at the end of the sentence in parenthesis like “…both in pregnant women and newborns (Table 2).” Same for Line 298. Avoid unnecessary wording.

12. Line 298: “As seen in Table 2, …There are also negative studies reported”. Could be rewritten as “there are positive and negative results using the probiotics in pregnancy and infancy.” Consider adding a new column in Table 2 to indicate the outcome.

13. Table 2 should be simplified, try to extract key information. I am wondering if it would read better by making another table with more specialized characteristics, such as probiotics used, single or mixture, test group feature (sex, age, predispositions), duration of the study, parameter records, positive/negative, reference, etc.

14. Simplify the conclusion section to focus on summarizing the review article (what is the main takeaway) and proposing future research directions.

15. Some of the writing in the conclusion section could be moved to the introduction. i.e., Line 421-425, “Probiotics are appealing to the public…requiring a prescription in the United States”.

Author Response

Reviewer 3

My comments are underlined HLD

Comments and Suggestions for Authors

Specific Comments

  1. Line 13: simplify the Background and Conclusion sections into one sentence, respectively (remove the term “background” and “conclusion”; expand and elaborate the Finding section (remove the term “Findings”)

I moved the second line from Background to Results making introduction one sentence long as you suggest. Secondly, I changed Findings to Results following instructions of the journal. I left the conclusions with two sentences since they both are the essence of the article.

  1. Employ another way to highlight your subheading, such as Line 77, 82, 130.

This was an important suggestion. I have bolded all major subheadings and indented subheadings and underlined these. Hopefully, it is now clear the heading to which these subheadings relate.

  1. Subheadingsfrom different levels should be distinct and easily recognizable.

I have redone these. Thank you for pointing this out to me.

  1. Only the “Introduction” (Line 31) and “The Microbiome during Pregnancy” (Line 47) are bolded. Maintain uniform formatting for all levels of headings.

This is now fixed.

  1. In Table 1, “Infant Microbiome Finding” should be bolded as well. Try to simplify the description using bullet points.

Terrific suggestions. The Table is improved.

  1. Would it read smoother if position mother-infant comparison after Line 193 (when you have discussed both sides).

I purposely put Mother-Infant pattern after talking about hygiene and antibiotic use in both groups and other factors influencing their microbiomes and at the end of that section as an introduction to summary of factors affecting them both listed in Table 1 that followed.

  1. Categorize Line 266 to 392 under “current microbiome treatments.” 304 to 430

Lines 274-368 of old proofs you had are under New subheading Current Microbiota Therapy… Thank you for suggestion.

  1. It would be helpful for the audience if there is another table/figure highlighting the disparities between previous and current/future probiotics treatment.

Thank you for this. I have redone Table 2. I believe I accomplished what you wanted. Also, I have tried to make it clear that the probiotics are all established probiotics, not new or advanced probiotics that are discussed later in the paper.

  1. Enhance the conciseness of Figure 1 by using visual aids and bullet points. Avoid citations in the figure.

Thank you, this looks better.

  1. Could you provide a brief description every time you introduce a new abbreviation? First example in table 2 under “Pregnant Women”, what does VSL#3 stand for? Also, Line 457, what is VOWST?

VSL#3 is the name used in 40 countries for an 8 probiotic combination preparation. The trade name has recently been changed to Visbiome. I have clarified these drugs in the new version of the paper and cited a reference for VOWST.

  1. Line 289 “As seen in Table 2, …” could be added at the end of the sentence in parenthesis like “…both in pregnant women and newborns (Table 2).” Same for Line 298. Avoid unnecessary wording.

I made this comment as you suggest in the text and altered Table 2 to more clear.

  1. Line 298: “As seen in Table 2, …There are also negative studies reported”. Could be rewritten as “there are positive and negative results using the probiotics in pregnancy and infancy.” Consider adding a new column in Table 2 to indicate the outcome.

Done.

  1. Table 2 should be simplified, try to extract key information. I am wondering if it would read better by making another table with more specialized characteristics, such as probiotics used, single or mixture, test group feature (sex, age, predispositions), duration of the study, parameter records, positive/negative, reference, etc.

This table has been redone to list the probiotics employed and to indicate if any of the areas included multiple studies. Mixture of probiotic strains is how they are used. It is rare for a single probiotic strain to be used.

  1. Simplify the conclusion section to focus on summarizing the review article (what is the main takeaway) and proposing future research directions.

Done

  1. Some of the writing in the conclusion section could be moved to the introduction. i.e., Line 421-425, “Probiotics are appealing to the public…requiring a prescription in the United States”.

Done

Thank you for your careful review of our paper and for your very helpful suggestions. I am sorry tracking changes did not work well on the manuscript which was changed quite a bit. 

Round 2

Reviewer 3 Report

Comments and Suggestions for Authors

The authors have formatted the article with improved overall structure. The table is now more straightforward and key points are highlighted. It would be much easier for the reviewing process if the author could use “track changes” in the updated manuscript.